# OBJECT FUSION VIA DIFFUSION TIME-STEP FOR CUSTOMIZED IMAGE EDITING WITH SINGLE EXAMPLE

## ABSTRACT

We tackle the task of customized image editing using a text-conditioned Diffusion Model (DM). The goal is to fuse the subject in a reference image (*e.g.*, sunglasses) with a source one (*e.g.*, a boy), while retaining the fidelity of them both (*e.g.*, the boy wearing the sunglasses). An intuitive approach, called LoRA fusion, first separately trains a DM LoRA for each image to encode its details. Then the two LoRAs are linearly combined by a weight to generate a fused image. Unfortunately, even through careful grid search or learning the weight, this approach still trades off the fidelity of one image against the other. We point out that the evil lies in the overlooked role of diffusion time-step in the generation process, *i.e.*, a smaller time-step controls the generation of a more fine-grained attribute. For example, a large LoRA weight for the source may help preserve its fine-grained details (*e.g.*, face attributes) at a small time-step, but could overpower the reference subject LoRA and lose the fidelity of its overall shape at a larger time-step. To address this deficiency, we propose *TimeFusion*, which learns a time-step-specific LoRA fusion weight that resolves the trade-off, *i.e.*, generating the source and reference subject in high fidelity given their respective prompt. Then we can customize image editing using this weight and a target prompt. Codes are in Appendix.

## 1 INTRODUCTION

Image editing modifies an image $I$ by altering user-defined visual attributes while retaining its fidelity, *i.e.*, preserving other attributes. For example, editing an image of a boy with the target prompt "wearing sunglasses" should only add sunglasses without altering the boy or his background. Recent efforts Hertz et al. (2022); Orgad et al. (2023); Tumanyan et al. (2023); Cao et al. (2023); Pan et al. (2023); Wallace et al. (2023); Wu & De la Torre (2023); Kawar et al. (2023); Hertz et al. (2023) utilize a text-conditioned Diffusion Model (DM) Ho et al. (2020), whose

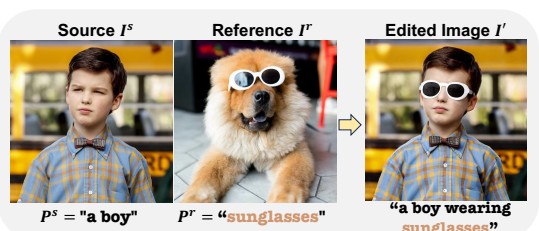

Figure 1: Customized image editing, which fuses the source image with the subject in a reference image, while retaining the fidelity of them both.

reverse process progressively transforms random noise into an image aligning with a given text prompt. The general paradigm involves two steps: first, calibrating the reverse process to reconstruct $I$ and retain fidelity; then, modifying it by introducing the target prompt to complete the edit. In particular, the main differences in existing methods lie in the reconstruction step. One approach, known as DDIM inversion Song et al. (2020), attempts to identify a noise initialization from which running the reverse process yields $I$. However, the inversion is prone to errors, leading to fidelity loss Tumanyan et al. (2023); Wallace et al. (2023), *e.g.*, distorting the appearance of the boy. Therefore, we base our study on an improved technique Hu et al. (2021) that involves learning a set of Low-Rank Adaptation (LoRA) layers injected into the DM (or learning a LoRA for short), which enables the LoRA-guided reverse process to reconstruct $I$.

Most existing image editing methods are text-based, which oftentimes lack customizability. For example, as shown in Figure 1, a user may want to edit a source image $I^s$ so that the boy wears the exact sunglasses in a reference image $I^r$ to produce the edited $I'$. In this case, it is impractical to

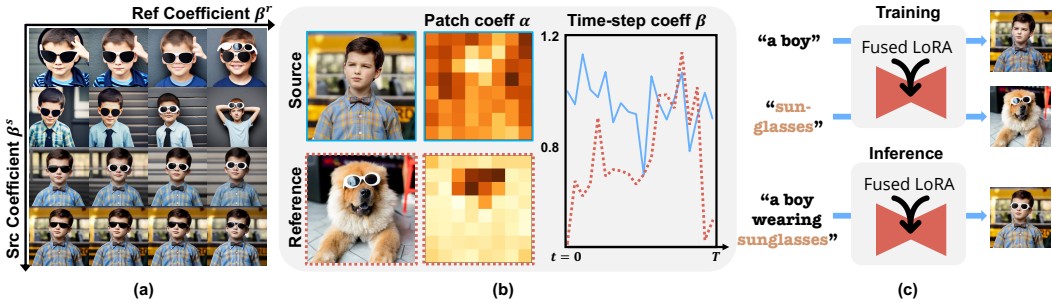

Figure 2: (a) Failure of current LoRA fusion in customized image editing, where no combination of coefficients maintains the fidelity of both images. (b) The proposed learnable time-step-specific coefficient and patch-specific one (visualized values are averaged across the LoRA injection layers). (c) The high-level training and inference pipeline for the proposed TimeFusion.

fully specify the appearance of the sunglasses using textual control. We aim to bridge this gap by exploring **customized image editing**, where the goal is to fuse the subject in $I^r$[1] with an image $I^s$, while retaining the fidelity of both.

To address this, a natural extension to the above editing paradigm is LoRA fusion Ryu (2023): first, learn a source and reference LoRA to reconstruct $I^s$ and the subject in $I^r$, respectively; then linearly combine their guidance in the reverse process using a fusion coefficient $(\beta^s, \beta^r)$, *e.g.*, a larger $\beta^s$ increases the guidance strength from the source LoRA. However, as shown in Figure 2(a), no combination of coefficients maintains the fidelity of both $I^s$ and $I^r$, *i.e.*, increasing $\beta^r$ preserves the visual attributes of the reference subject but sacrifices the fidelity of $I^s$.

Therefore, the crux of customized image editing lies in finding a more precise method for fusing LoRAs that accommodates the visual attributes of both $I^s$ and $I^r$. This motivates us to improve LoRA fusion by considering the diffusion time-step (along which the reverse process generates an image). This is based on two key observations: 1) Diffusion time-step is theoretically and empirically linked to visual attributes Yue et al. (2024a;b), where the reverse process at a smaller time-step is responsible for generating a more fine-grained attribute. For example, as we will later show in Figure 4, providing LoRA guidance at smaller time-steps modifies more fine-grained attributes. 2) Thus, a time-step-specific fusion coefficient provides the required precision, *e.g.*, using a large $\beta^s$ at a small time-step to preserve the fine-grained facial details in $I^s$, while increasing $\beta^r$ at a larger time-step to maintain the more coarse-grained shape of the sunglasses in $I^r$ (more examples in Figure 13).

Building on the above analysis, we propose a time-step specific LoRA fusion strategy called **TimeFusion**. Specifically, we first learn a source and reference LoRA using a standard technique Avrahami et al. (2023) (preliminaries provided in Appendix). Then our coefficients for LoRA fusion consist of two parts, as shown in Figure 2(b): 1) the aforementioned time-step specific ones $(\beta_t^s, \beta_t^r)$ for each diffusion time-step $t \in \{1, \ldots, T\}$. 2) a patch-level one $\boldsymbol{\alpha}^s, \boldsymbol{\alpha}^r \in \mathbb{R}^{8 \times 8}$ for the $8 \times 8$ latent patches in each layer where LoRA is injected (layer index omitted for simplicity). The patch-level coefficient acts as a modifier to the time-step one, further refining the fusion by considering spatial information, *e.g.*, applying a large source LoRA coefficient on a background patch to faithfully reconstruct $I^s$. Overall, during the reverse process at time-step $t$, the LoRA fusion coefficient for a patch at spatial location $(i, j)$ will be $(\boldsymbol{\alpha}_{i,j}^s \beta_t^s, \boldsymbol{\alpha}_{i,j}^r \beta_t^r)$. The paradigm of TimeFusion is summarized in Figure 2(c). We train the coefficients so that the fused LoRA reconstructs $I^s$ and $I^r$ according to their respective prompt, *i.e.*, accommodating the visual attributes of both images. In inference, using the learned LoRA fusion, we achieve customized image editing by simply supplying the target prompt. Figure 3 showcases our results. Our contributions include:

- We tackle the challenging task of customized image editing with a text-conditioned DM (Section 2.1), by improving the current LoRA fusion (Section 2.2).

- Motivated by the connection between diffusion time-step and visual attribute (Section 2.1), we propose TimeFusion, a novel time-step-specific LoRA fusion strategy in Section 3.

---

[1] We segment the subject with SAM Kirillov et al. (2023), requiring the user to click its location in $I^r$ based on the prompt $P^r$. Note that this can be easily automated with a text-conditioned segmentation model such as Rasheed et al. (2024).

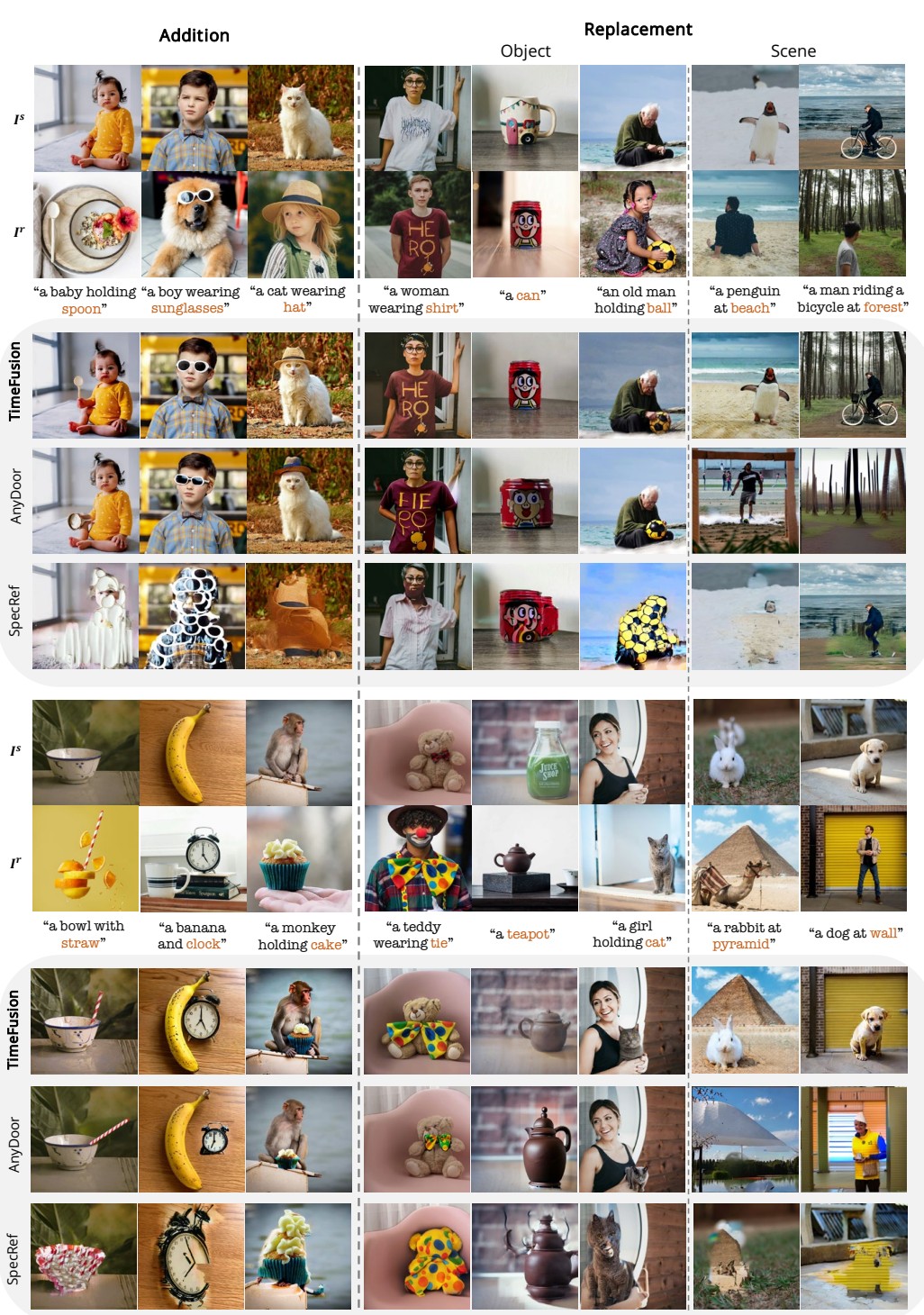

Figure 3: Comparison of customized image editing results between our TimeFusion and existing SOTAs Chen et al. (2023); Chen & Huang (2023). We compare 3 tasks, including addition, object and scene replacement (details in Section 3). The source prompt is omitted, and the reference subject prompt is highlighted in orange inside the target prompt. For fairness, examples are chosen based on their best visual quality from various random seeds. See Section 5 for analysis and Appendix for additional results.

- Extensive qualitative and quantitative experiments in Section 5 demonstrate the superiority of our TimeFusion over existing works in customized image editing.

## 2 PROBLEM FORMULATIONS

### 2.1 TEXT-CONDITIONED DIFFUSION MODEL (DM)

A text-conditioned DM is a generative model that allows textual control. It uses a *forward process* that incrementally adds noise to input data to learn a *reverse process*, where the model is trained to reconstruct clean data from noisy one and its text description. In this work, we focus on Stable Diffusion (SD) Rombach et al. (2022), where each input data $\mathbf{z}_0$ is an image feature. Specifically, $\mathbf{z}_0$ is the latent variable of a pre-trained image autoencoder, which is encoded from an image to reconstruct it. Hence we use $\mathbf{z}_0$ and image interchangeably when the context is clear.

**Forward Process**. It progressively adds Gaussian noise to each image $\mathbf{z}_0$ in $T$ time-steps, producing noisy images $\mathbf{z}_1, \ldots, \mathbf{z}_T$, with the subscript denoting time-step. Each $\mathbf{z}_t$ adheres to its noisy image distribution $q(\mathbf{z}_t|\mathbf{z}_0)$, which gradually collapses towards a pure noise with reducing mean and increasing variance as $t$ increases (details in Appendix). Recent works Yue et al. (2024a;b) show that the forward process connects diffusion time-step with the visual attributes of an image. In a nutshell, an increasing $t$ causes a large overlap between noisy image distributions, essentially collapsing different images into similar ones by losing the visual attributes that differentiate them. In particular, the theory suggests that fine-grained attributes (*i.e.*, those affecting local appearances, such as expression) become lost at a smaller time-step compared to coarse-grained ones (*i.e.*, those affecting global appearances, such as background). This pattern of attribute loss has interesting implications in the following reverse process.

**Reverse Process**. It corresponds to a learned Gaussian transition $p_\theta(\mathbf{z}_{t-1}|\mathbf{z}_t, P)$ conditioned on a text prompt $P$ and parameterized by $\theta$. The term is computed in two steps: first reconstructing $\mathbf{z}_0$ as $\mathbf{z}_0'$ by a learnable denoising network $d(\mathbf{z}_t, P, t; \theta)$ with parameter $\theta$, then computing $q(\mathbf{z}_{t-1}|\mathbf{z}_t, \mathbf{z}_0')$, which has a closed-form solution given in Appendix. In training, we minimize the reconstruction error:

$$\mathcal{L}(\theta, \mathbf{z}_0, P) = \sum_{t=1}^{T} \mathbb{E}_{q(\mathbf{z}_t|\mathbf{z}_0)} \|\mathbf{z}_0 - d(\mathbf{z}_t, P, t; \theta)\|^2, \tag{1}$$

where $\mathbf{z}_t$ is sampled from the forward process at a random time-step $t$. In inference, to generate an image conditioned on a prompt $P$, we first set $\mathbf{z}_T$ as a random noise, and recursively run the reverse process $p_\theta(\mathbf{z}_{t-1}|\mathbf{z}_t, P)$ until obtaining $\mathbf{z}_0$. Then for SD, it has a decoder that maps $\mathbf{z}_0$ to an image in the pixel space. In particular, due to the progressive loss of attributes going from $\mathbf{z}_0$ to $\mathbf{z}_T$ in the forward process, the reverse process must correspondingly make up for the lost attribute in each step to accurately reconstruct $\mathbf{z}_0$. Hence as fine-grained attributes are lost at smaller time-steps, the reverse process is responsible for generating them at smaller time-steps correspondingly. The technique introduced in the next section will help us visualize this.

### 2.2 LOW-RANK ADAPTATION (LORA)

Training a DM with a randomly initialized $\theta$ by minimizing Equation 1 can be extremely expensive. Hence the common approach is to initialize $\theta$ from a DM pre-trained on diverse image data (*e.g.*, SD), and fine-tune it to a downstream task. LoRA is the most mainstream fine-tuning method Hu et al. (2021); Song et al. (2024); Gu et al. (2024).

LoRA refers to a set of low-rank matrices, each of which is injected into a corresponding layer of DM to update its weight. Specifically, let $W \in \mathbb{R}^{m \times n}$ denote the pre-trained weight matrix of a layer in DM. The LoRA matrice injected to this layer is given by $\Delta W = AB$, where $A \in \mathbb{R}^{m \times r}, B \in \mathbb{R}^{r \times n}$, such that the rank of $\Delta W$ equals a small $r \ll \min(m, n)$. After injection, the weight of this layer becomes $W + \Delta W$.

**LoRA Training**. In fine-tuning, the original DM weights are frozen, and only the injected LoRA is trained. We denote the LoRA matrices as $\theta^l$ (or LoRA $\theta^l$ for short), and the weight of DM after LoRA injection as $\theta \oplus \theta^l$. Given a downstream dataset $\mathcal{D}$ where each sample $(\mathbf{z}_0, P)$ comprises of

Figure 4: Effect of applying LoRA guidance at a subset of diffusion time-steps (highlighted by a red bar), ranging from no guidance in $I_0$ to full guidance in $I_4$. The LoRA is learned to reconstruct $I$ given $P$. All images are generated with the same noise initialization.

an image $\mathbf{z}_0$ and its prompt description $P$, the objective of fine-tuning is given below:

$$\min_{\theta^l} \sum_{(\mathbf{z}_0, P) \in \mathcal{D}} \mathcal{L}(\theta \oplus \theta^l, \mathbf{z}_0, P). \tag{2}$$

**LoRA in Image Editing**. This is a special case of the above fine-tuning process, where $\mathcal{D}$ contains only the image $I$ for editing and its prompt $P$ (*e.g.*, "a boy"), *i.e.*, we essentially learn a LoRA $\theta^l$ to reconstruct $I$. To generate an edited image, one can run this reverse process parameterized by $\theta \oplus \theta^l$ with a modified prompt $P'$ (*e.g.*, "a smiling boy" to change his expression). In particular, we can visualize the effect of LoRA guidance (by $\Delta W$) at different ranges of time-steps. In Figure 4, we compare the image generated by the original SD ($I_0$), by only injecting LoRA at a subset of time-steps ($I_1, I_2, I_3$) and by injecting LoRA at all time-steps ($I_4$). It is clear that the guidance controls a more fine-grained attribute at a smaller time-step, *e.g.*, $I_3$ (guiding large time-steps) mainly retains the overall background of $I$, while $I_1$ (guiding small ones) mainly alters the local face attributes from $I_0$.

**LoRA Fusion**. Without loss of generality, one can fuse two LoRAs trained on different datasets by a tuple of tunable strength coefficients $(\beta_1, \beta_2)$. After injecting the fused LoRAs, the weight of a DM layer becomes $W + \beta_1 \Delta W_1 + \beta_2 \Delta W_2$, where $\Delta W_i$ and $\beta_i$ denote the corresponding low-rank matrix in the $i$-th LoRA and its coefficient, respectively. An example use case of LoRA fusion is synthesizing a subject in a specific style Shah et al. (2023), where a LoRA trained on images of a subject (*e.g.*, one specific dog) is fused with the other LoRA on one image of a style (*e.g.*, watercolor). However, as we analyze in the introduction, the current way of fusing LoRAs does not have the required precision to tackle the customized image editing task (Figure 2(a)). In the next section, we propose TimeFusion to fix this.

## 3 TimeFusion

We aim to tackle the customized image editing task: given a source image $I^s$ and its text prompt $P^s$, a reference image $I^r$ containing a subject described by a prompt $P^r$, the goal is to fuse the subject in $I^r$ with $I^s$ according to a target prompt $P'$, while retaining their fidelity.

Our TimeFusion is an extension to LoRA fusion discussed in Section 2.2, where the fusion coefficient additionally depends on diffusion time-step and spatial location in the feature map. It consists of three steps: 1) learn a LoRA to reconstruct $I^s$ and $I^r$, respectively; 2) initialize time-step-specific coefficients and patch-specific coefficients for LoRA fusion; 3) learn the coefficients to retain the reconstruction capability of each individual LoRA after fusing them. We detail each step below:

**Step 1**. We aim to learn a LoRA $\theta^s$ to reconstruct $I^s$, and a LoRA $\theta^r$ to reconstruct the subject in $I^r$. For pre-processing, we use SAM Kirillov et al. (2023) to get the subject mask in $I^r$ by asking the user to click the subject location based on $P^r$. We leave it as future work to automate this with a text-conditioned segmentation model, *e.g.*, by combining SAM with Grounding DINO Liu et al. (2023). After getting the subject mask, we train the LoRAs by:

$$\min_{\theta^s} \mathcal{L}(\theta \oplus \theta^s, \mathbf{z}_0^s, P^s), \qquad \min_{\theta^r, [\text{V}]} \mathcal{L}(\theta \oplus \theta^r, \hat{\mathbf{z}}_0^r, \text{"[V]"}), \tag{3}$$

where $\theta$ is the pre-trained weight of SD, $\mathbf{z}_0^s$ denotes the image latent of $I^s$, $\hat{\mathbf{z}}_0^r$ denotes the image latent of $I^r$ after applying the subject mask (*i.e.*, $\mathcal{L}$ is only evaluated inside the mask), and [V]

Figure 5: The overall pipeline of learning the time-step-specific and patch-specific coefficients (Steps 2 and 3 in Section 3). We highlight the reference object (segmented by SAM) with a blue border. The lock and unlock icon denotes frozen and trainable parameters, respectively. On the right, we detail our LoRA fusion strategy on an example layer in the SD U-Net, where $\times$ denotes the element-wise product. We omit the channel dimensions in layer input and output for simplicity.

denotes a learnable token embedding following standard practice Avrahami et al. (2023); Gal et al. (2022). We slightly abuse the notation to put [V] inside the prompt.

**Step 2.** For *time-step-specific coefficients*, we define $\beta_t^s, \beta_t^r$ for each $t \in \{1, \ldots, T\}$. In practice, we find it unnecessary to learn a unique coefficient at each $t$. Instead, we sequentially group the time-steps into $K$ splits of equal size (*e.g.*, the first split being $1, \ldots, T/K$), and share the coefficient value inside each split. We ablation the effects of $K$ in Figure 11. For *patch-specific coefficients*, we define $\alpha^s, \alpha^r \in \mathbb{R}^{8 \times 8}$ for each layer with LoRA injection (layer index omitted for simplicity). Note that the spatial dimension of the feature map at different layers in SD can be different, *i.e.*, ranging from 8×8 to 64×64. To keep our coefficients simple, we fix the number of patches to $8 \times 8$ by varying the patch size at different layers (*e.g.*, 8 when the spatial dimension is $64 \times 64$). Overall, at time-step $t$, the LoRA fusion coefficient for a patch at location $(i, j)$ (out of the $8 \times 8$ patches) is $(\alpha_{i,j}^s \beta_t^s, \alpha_{i,j}^r \beta_t^r)$, as shown in Figure 5 right. We denote the set of all coefficients as $\boldsymbol{\beta}$, and the DM weight after injecting the fused LoRA as $\theta \oplus \theta^{\boldsymbol{\beta}}$. We initialize all coefficients in $\boldsymbol{\beta}$ as 1 and train them in the next step.

**Step 3.** As illustrated in Figure 5 left, we learn $\boldsymbol{\beta}$ by the following objective:

$$\min_{\boldsymbol{\beta}} \mathcal{L}\left(\theta \oplus \theta^{\boldsymbol{\beta}}, \bar{\mathbf{z}}_0^s, P^s\right) + \mathcal{L}\left(\theta \oplus \theta^{\boldsymbol{\beta}}, \bar{\mathbf{z}}_0^r, \text{``[V]''}\right),  \qquad (4)$$

where [V] is the token embedding learned in Step 1, $\bar{\mathbf{z}}_0^s, \bar{\mathbf{z}}_0^r$ denotes the reconstructed image latent by LoRA $\theta^s$ and $\theta^r$, respectively. Overall, we are training the fusion weight, such that the fused LoRAs can accommodate all visual attributes of $I^s$ and $I^r$ to accurately reconstruct their latents.

**Generating Edited Image.** After training, we replace the subject name in the target prompt $P'$ by the learned token [V] (*e.g.*, "a boy wearing sunglasses"→"a boy wearing [V]" for subject "sunglasses"). Then we run the reverse process parameterized by $\theta \oplus \theta^{\boldsymbol{\beta}}$ given the modified $P'$ to get the edited image. We highlight two points: 1) This scheme can be extended to do object replacement (*e.g.*, replacing a mug with "can" by $P' = $ "[V]" with [V] being the learned token for "can") or scene replacement (*e.g.*, changing a "dog" background by $P' = $ "a dog at [V]" with [V] being the learned token for a background). 2) Once the LoRAs and their fusion coefficients are learned, they can generalize to different editing prompts without additional fine-tuning, leveraging the prior knowledge of SD (Appendix Figure 12).

## 4 RELATED WORK

**Image Editing** are mostly Text-Based (TBIE). To maintain the fidelity of the user-provided image, a line of works Cao et al. (2023); Tumanyan et al. (2023) adopt DDIM inversion Song et al. (2020) or its enhanced versions Pan et al. (2023); Wallace et al. (2023) for accelerated reconstruction. However, such methods are unstable and prone to errors. To improve the reconstruction, fine-tuning SD or injected LoRA layers Kawar et al. (2023); Zhang et al. (2023) provides a promising solution. Besides reconstruction, several techniques are proposed to improve editing, *e.g.*, by adjusting attention maps Hertz et al. (2022), interpolating learnable text embeddings Kawar et al. (2023) and

encoding semantic change in the text LoRA Song et al. (2024). However, customized image editing is under-explored. The most relevant work SpecRef Chen & Huang (2023) is based on DDIM inversion, where the reference image feature is extracted in the inversion stage, and incorporated when adjusting attention maps for editing. Yet its performance is bottlenecked by DDIM inversion. We improve it by extending LoRA fusion for customized image editing.

**LoRA Fusion** is mainly used for multi-concept customization, which synthesizes novel combinations of concepts, without the need to perverse the fidelity of any user-provided images. Existing works focus on improving LoRA training Po et al. (2023) or fusion strategy Zhong et al. (2024). For the latter, some notable improvements include Mix-of-Show Gu et al. (2024), which proposes gradient fusion that directly learns the combined LoRA weights without losing the capability for single concept generation; MoLE Wu et al. (2023a), which leverages learnable gating functions for each LoRA layer to fulfill the combination; and ZipLoRA Shah et al. (2023), which claims that the columns of two LoRA weights should be orthogonal to avoid identity loss and optimizes orthogonal coefficients for each column of two LoRAs. However, we find existing LoRA fusion strategy lacks the required precision in customized image editing. Hence we extend it with time-step-specific and patch-specific fusion coefficients.

## 5 EXPERIMENT

**Implementation Details.** Following prior editing works Kawar et al. (2023); Song et al. (2024); Zhang et al. (2023), we adopt Stable Diffusion Rombach et al. (2022) as our DM and use its default parameters. When learning LoRA $\theta^s$ to reconstruct $I^s$, the learning rate is set as 1e-4 and the optimization iteration is 800. We use the approach in Avrahami et al. (2023) to learn the subject in a single image $I^r$. For time-step-specific coefficients, we equally divide the time-steps into 20 splits, *i.e.*, $K = 20$. When learning with Equation 4, we set the learning rate as $5e^{-2}$ and training iterations as 100. Experiments are conducted on an NVIDIA A100 GPU with a batch size of 1. Computation analysis is included in Appendix.

**Dataset.** To evaluate the effectiveness of our TimeFusion in handling various objects, we collected images separately as source and reference ones from the widely used website, *i.e.*, Unsplash (https://unsplash.com/). In particular, the number of source images is 18 and the main subjects include humans, animals, and objects. For the reference images, there are objects, animals, and scenes totaling 20 images. During the fusion of source and reference images, each source image is paired with 2-3 different reference images. Finally, we obtain 50 source-reference pairs with corresponding prompts for customized image editing, including 20, 18, and 12 samples for object addition, object replacement, and scene replacement, respectively.

### 5.1 QUALITATIVE EVALUATION

**Our Results.** Our proposed TimeFusion supports both addition and replacement editing. Especially, the replacement involves object and scene replacement. The qualitative results are shown in Figure 3 and more results are in Appendix. Overall, our generated images largely preserve the fidelity of both the source images and the reference subjects while achieving high alignment with target prompts, demonstrating the superiority of TimeFusion's fusion and editing capability. For example, we could make a baby holding a spoon and replace a bottle with a teapot. We highlight the robustness of

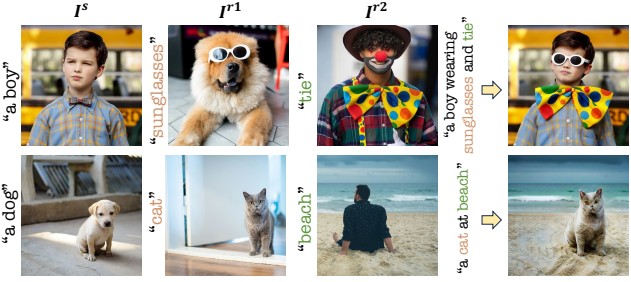

Figure 6: Customized image editing with three objects.

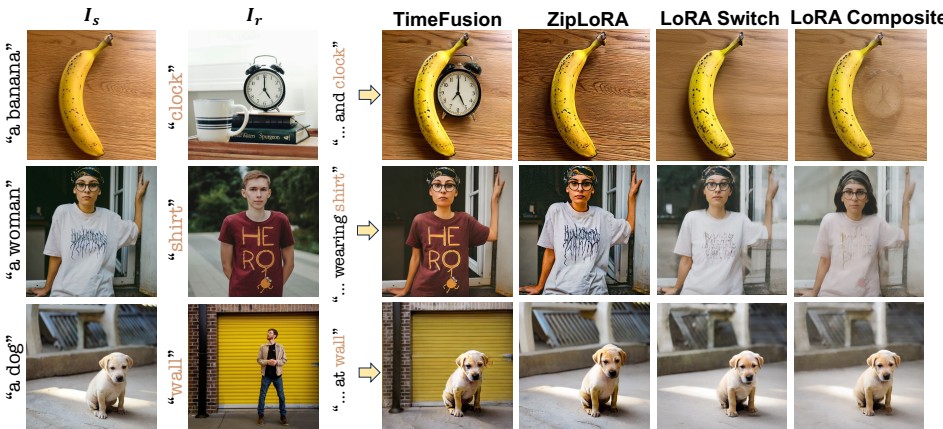

Figure 7: Qualitative comparison with LoRA fusion methods.

our TimeFusion from three perspectives: 1) Our method supports significant changes in the object's overall posture, *e.g.*, making the cat from standing to sitting, being held, and jumping in Figure 14. 2) We also support multiple reference images, *e.g.*, making the boy wear the sunglasses and tie, and changing the dog to the cat at the beach (while retaining the position and posture of the dog in $I^s$) in Figure 6. 3) We are robust to a coarse reference subject mask, *e.g.*, a dilated mask in Figure 15.

**Comparison with SOTAs.** The comparison is in Figures 3, 16 and 17. The closest work SpecRef Chen & Huang (2023) is based on DDIM inversion. Yet it falls short in customized image editing due to the error-prone inversion process. AnyDoor Chen et al. (2023) adopts a more restrictive setting. It additionally requires the user to provide the editing mask in the source image for inserting the reference subject, *i.e.*, the model can conveniently edit the user-provided area with other parts fixed. Even under this easier task, we observe that it often loses the reference subject fidelity (*e.g.*, the sunglasses, hat, or the can in the first row of Figure 3). Furthermore, some fused subjects look unnatural in the source image (*e.g.*, the clock or the cat in the last row of Figure 3). We also highlight a limitation of AnyDoor's setting. While a user-provided editing mask reduces the complexity of the task, it may accidentally lead to inferior results in some cases, *e.g.*, changing a cup to a can in the first row of Figure 3 will require editing the reflection on the table, but this can be easily overlooked by the user when providing the mask, causing the reflection unedited.

**Comparison with Multi-Concept Customization Methods**. The other close line of works Shah et al. (2023); Zhong et al. (2024) aims to achieve multi-concept customization through LoRA fusion. We compare with them in Figure 7, where none of these baselines achieves successful editing. They either make almost no edit to the source image (*e.g.*, ZipLoRA Shah et al. (2023)) or fail to fuse the reference subject correctly (*e.g.*, LoRA Switch and LoRA Composite Zhong et al. (2024)). In contrast, our TimeFusion produces high-quality editing results, showing that it provides the required precision to preserve the fidelity of both the source images and reference subjects.

Table 1: Quantitative evaluation of our TimeFusion against SOTAs. See main text for metric explanation.

|  | CLIP-T ↑ | CLIP-I ↑ | DINO ↑ |
|---|---|---|---|
| SpecRef | 0.266 | 0.694 | 0.778 |
| AnyDoor | 0.288 | 0.780 | 0.780 |
| TimeFusion | 0.316 | 0.828 | 0.806 |

Table 2: Ablation of using only time-step-specific coefficients, only patch-specific ones, and standard LoRA fusion with a single set of coefficients.

| Method | $\beta_t$ | $\alpha$ | CLIP-T ↑ | CLIP-I ↑ | DINO ↑ |
|---|---|---|---|---|---|
| Time-step only | ✓ |  | 0.316 | 0.820 | 0.795 |
| Patch only |  | ✓ | 0.297 | 0.871 | 0.768 |
| LoRA fusion |  |  | 0.321 | 0.797 | 0.788 |
| TimeFusion | ✓ | ✓ | 0.316 | 0.828 | 0.806 |

## 5.2 QUANTITATIVE EVALUATION

**Similarity Metrics.** Since customized image editing requires image-level alignment with the source image and subject-level alignment with the reference subject, we consider the metrics in both con-

cept customization Gu et al. (2024); Shah et al. (2023) and image editing Kawar et al. (2023); Song et al. (2024): 1) text alignment (CLIP-T) which measures the CLIP similarity between the target prompt and the edited image; 2) source alignment (CLIP-I) which measures the CLIP similarity between the source image and edited image; 3) reference alignment (DINO), which computes the cosine similarity between ViT S/16 DINO embeddings Caron et al. (2021) of reference and edited images. It is worth noting that as SD generates an image that matches the CLIP text embedding of the given prompt, the direct text-to-image results from SD (with no control) will have the highest CLIP-T score. Moreover, making no edits to the source image will lead to the highest CLIP-I score. However, none of these two situations are expected in customized image editing. Therefore, the key is to obtain a balanced CLIP-I and CLIP-T score, instead of having a high score in only one of them. The results are summarized in Table 1, where our TimeFusion achieves the best scores across the three metrics compared with SpecRef and AnyDoor.

**User Study.** We further evaluate our proposed TimeFusion through an extensive human perceptual evaluation study. It consists of 50 source-reference pairs with corresponding prompts. 51 AMT evaluators participated in this study to rate the editing quality of the 50 samples. Each sample includes a source image, a reference image containing a specified subject, a target prompt, and three edited images generated by TimeFusion, AnyDoor, and SpecRef, which are randomly shuffled. They are required to choose the best result among the three. Finally, we collected 2,550 answers and the results are depicted in Figure 8, where 89.6% of participants prefer our TimeFusion.

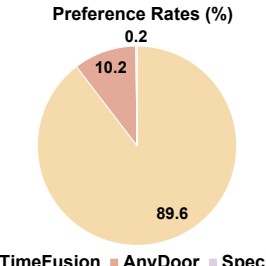

Figure 8: User study.

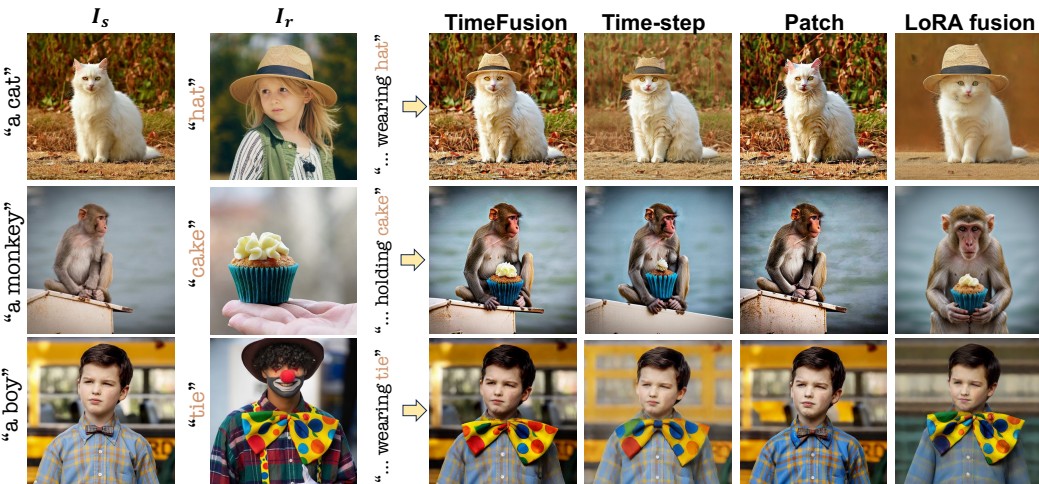

Figure 9: Qualitative comparison by ablating the use of fusion coefficients. "Time-step" and "Patch" denote using only the corresponding coefficients. "LoRA fusion" denotes its standard approach with a single set of coefficients (Section 2.2).

### 5.3 ABLATION ANALYSIS

**Ablation on LoRA Coefficients.** We ablation the use of time-step-specific coefficient and patch-specific one in Figure 9 and Table 2. The "LoRA fusion" coefficients are obtained by grid-search, and the rest are learned by Equation 4. With only time-step coefficients, we observe a loss of spatial information, *e.g.*, the image backgrounds of the cat and boy are blurry, and the cake held by the monkey loses its details. Only using patch coefficients barely makes any edit in Figure 9, hence it is the best in CLIP-I but the worst in the other two metrics in Table 2. The LoRA fusion has little control (only a single set of coefficients) of the generated image, hence it has the highest CLIP-T score, yet alters the source image significantly (lowest CLIP-I score). Finally, since time-step coefficient $\beta_t$ provides the fusion precision and patch coefficient $\alpha$ refines spatial information, our TimeFusion equipped with both coefficients gains a balanced score as expected in the above "Similarity Metrics" and achieves the best results.

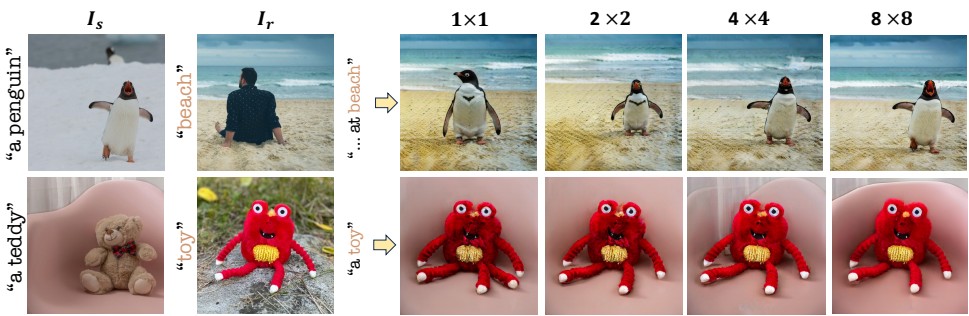

Figure 10: Ablation on the size of patch-level coefficients. TimeFusion uses $8 \times 8$ (Section 3).

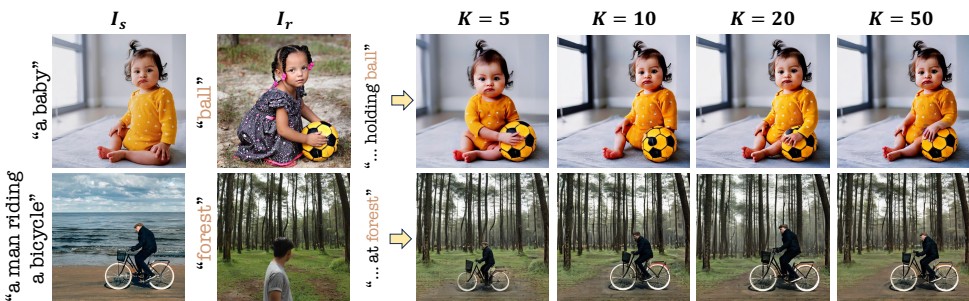

Figure 11: Ablation on the number of splits $K$ for the time-step-specific coefficients.

**Ablation on Patch-level Coefficient.** We examine the effect of the patch-level coefficient's size on the editing results in Figure 10. It could be observed that a small size provides only coarse spatial information, hence it fails to maintain the fidelity of either the source image or reference subject, *e.g.*, the posture and position of the penguin and the appearance of the toy are not well preserved. As the size of the patch-level coefficient increases from $1 \times 1$ to $8 \times 8$, the generated images progressively achieve better fidelity. Therefore, we used $8 \times 8$ in TimeFusion.

**Ablation on Time-step Coefficient.** We ablation the number of splits $K$ for the time-step coefficient. As depicted in Figure 11, as $K$ increases from 5 to 20, we observe consistent improvements in fidelity, *e.g.*, the baby's posture at $K = 5$ and the baby's leg at $K = 10$ are not preserved. This is because when $K$ is small, a long range of time-steps shares the same time-step coefficients, yet corresponds to multiple visual attributes. Hence the fusion may be imprecise to lose fidelity. However, further improving $K$ to 50 leads to a slight loss of fidelity, *e.g.*, the shade of the baby's clothes. We conjecture that the reason is inadequate learning for coefficients: as the optimization iteration of fusion is fixed to 100, each time-step coefficient is only expected to be trained twice when $K = 50$. Yet increasing the iteration increases the training time. Hence we choose $K = 20$ to balance the editing results and compute.

## 6    CONCLUSION

We presented TimeFusion, which enables customized image editing with the text-conditioned Stable Diffusion (SD). The crux is to first learn two SD LoRAs for encoding the visual attributes of the source image and reference subject, respectively, and then fuse them in a precise way to maintain the fidelity of both. Motivated by the connection between diffusion time-step and visual attributes, we extend the current LoRA fusion with learnable time-step-specific and patch-specific coefficients. They are trained such that SD, after LoRA fusion, can still reconstruct the source image and reference subject. We show that the additional coefficients enable LoRA fusion to simultaneously accommodate the visual attributes of the user-provided source and reference images. Hence we significantly improve the fidelity in customized image editing, compared to the current state-of-the-arts. As a future direction, we will speed up the LoRA learning process in training and the SD reverse process in editing by exploring Fast Diffusion Model Wu et al. (2023b) and Consistency Models Song et al. (2023).

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

## A   APPENDIX / SUPPLEMENTAL MATERIAL

This Appendix is organized as follows:

- Section A.1 lists the limitations of TimeFusion.
- Section A.2 includes the computation analysis.
- Section A.3 analyses the generalization to different prompts.
- Section A.4 illustrates the details of learning source and reference LoRAs.
- Section A.5 provides the detailed formulation of DM's forward and reverse process.
- Section A.6 visualizes more time-step coefficients, varies object posture changes, discusses the effect of subject masks, and showcases additional results.

| Abbreviation/Symbol | Meaning |
|---|---|
| *Abbreviations* | |
| TBIE | Text-Based Image Editing |
| LoRA | Low-Rank Adaptation |
| DM | Diffusion Model |
| SD | Stable Diffusion |
| *Symbols in Method* | |
| $I^s, I^r$ | Source, reference image |
| $P^s, P^r$ | Source, reference subject prompt |
| $P', I'$ | target prompt, edited image |
| $\beta^s, \beta^r$ | LoRA fusion coefficient |
| $\beta_t^s, \beta_t^r$ | Time-step-specific coefficient |
| $\boldsymbol{\alpha}^s, \boldsymbol{\alpha}^r$ | Patch-specific coefficient |
| $\mathbf{z}$ | Image latent by SD encoder |
| $\mathbf{z}_1, \ldots, \mathbf{z}_T$ | Noisy samples |
| $\theta$ | SD weight |
| $\mathcal{D}$ | Downstream dataset |
| $\theta^s, \theta^r$ | Source, reference image LoRA |
| $\mathcal{L}$ | DM loss |
| [V] | Learnable subject token embedding |
| $\boldsymbol{\beta}$ | Set of all fusion coefficients |
| $\oplus$ | LoRA injection |

Table 3: List of abbreviations and symbols used in the paper.

### A.1   LIMITATIONS

Our approach performs sub-optimally when the target prompt contains orientations, *e.g.*, "in front of", due to Stable Diffusion's limitation. We also leverage the diffusion model for generation, whose reverse process can be expensive (compared to GAN's one-step generation).

### A.2   COMPUTATION ANALYSIS

The detailed computational resources and time consumption of TimeFusion and each compared method are listed as follows:

1. Tuning-free methods: AnyDoor takes 10-second DDIM sampling with 18G GPU VRAM. However, it often fails to preserve the fidelity of the reference subject (*e.g.*, the sunglasses, hat, or the can in the first row of Figure 3). Furthermore, some fused subjects look unnatural in the source image (*e.g.*, the clock or the cat in the last row of Figure 3).

2. DDIM inversion methods: SpecRef costs 29 seconds for inversion followed by DDIM sampling with 16G GPU VRAM. Yet due to the error-prone inversion process, it fails in object addition and has low fidelity in replacement manipulations (Figure 3).

Figure 12: Customized image editing with different prompts.

3. LoRA fusion methods: They all first train two LoRAs for the source image and reference subject which consumes around 6 minutes each, with 14G GPU VRAM. During fusing LoRA, our TimeFusion consumes 4 minutes with 20G GPU VRAM and ZipLoRA costs 6 minutes with 24G GPU VRAM. Serving as tuning-free approaches for LoRA combination, LoRA Switch and LoRA Composite take 29-second and 21-second DDIM sampling with both 8G GPU VRAM. However, ZipLoRA, LoRA Switch, and LoRA Composite struggle to fulfill a good trade-off to preserve the fidelity of the source image and reference subject simultaneously (Figure 7).

### A.3 GENERALIZATION ANALYSIS

Our TimeFusion adopts time-step-specific and patch-specific coefficients to fuse source image LoRA and reference subject LoRA. It is worth noting that these two coefficients are learned such that the fused LoRA can reconstruct the source image and the reference subject with their respective prompts, *i.e.*, not overfitting to any of the two images. Take the reference patch coefficient $\alpha^r$ for example, the reference subject having a large weight $\alpha^r$ on some positions does not imply that it is fixed to its layout and pose in the final generation. This is because the generation is affected not only by reference patch coefficient $\alpha^r$, but also by source patch coefficient $\alpha^s$ and time-step coefficients $\beta_t^s$ and $\beta_t^r$, *e.g.*, $\alpha^r$ does not affect source image reconstruction given source prompt. In fact, having both patch and time-step coefficients gives the model enough flexibility to accommodate source and reference image attributes, allowing the final generation to be faithful to the user's prompt.

Furthermore, in the LoRA fusion methods for multi-concept customization, the user prompt is not known a priori. To enable fair comparison with existing works, we choose to learn prompt-agnostic coefficients. To follow the user-specified prompt after fusion, we simply leverage SD's extrapolating power to imagine the composition of the source image and reference subject, which is empirically justified and extensively leveraged in any SD-based image editing works Kawar et al. (2023); Tumanyan et al. (2023). Actually, prompt-agnostic fusion has the added benefit of quickly adapting to different prompts given the same source and reference images. As shown in Figure 12, we could add the clock or replace the banana with the clock, and make the squirrel hold the candle or next to the candle.

### A.4 DETAILS ON LEARNING LORA

**Source LoRA.** We learn a LoRA $\theta^s$ to reconstruct the source image $I^s$. Then the weight of SD after injecting $\theta^s$ becomes $\theta \oplus \theta^s$, where $\theta$ is the pre-trained weight of SD. During training, we follow the objective function as formulated in Equation 3: $\min_{\theta^s} \mathcal{L}(\theta \oplus \theta^s, \mathbf{z}_0^s, P^s)$, where $\mathbf{z}_0^s$ and $P^s$ denote the image latent and text prompt of $I^s$ respectively, $\mathcal{L}$ is defined in Equation 1. After training, utilizing the SD parameterized by $\theta \oplus \theta^s$, we could input prompt $P^s$ to obtain reconstructed $I^s$.

**Reference LoRA.** Considering the LoRA $\theta^r$ for reconstructing a reference subject, we adopt the technique in Avrahami et al. (2023). Given a reference image $I^r$ containing a specified subject indicated by mask $M$ and prompt $P^r$ (*e.g.*, "sunglasses"), Avrahami et al. (2023) adapt the standard

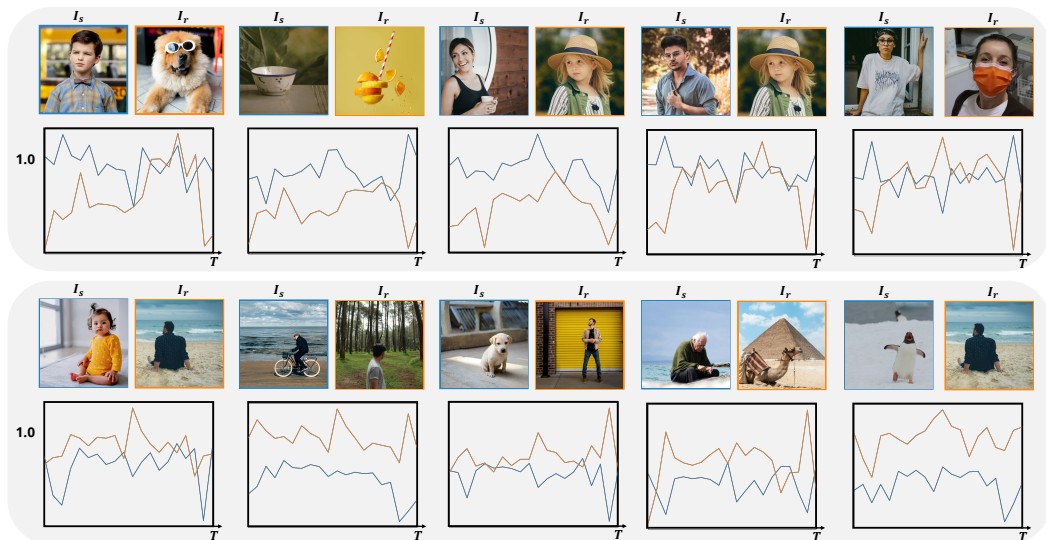

Figure 13: Time-step coefficient visualization.

diffusion loss in Equation 1 into a masked version:

$$\mathcal{L}_{mask}(\theta \oplus \theta^r, \mathbf{z}_0^r, \text{``[V]''}) = \sum_{t=1}^{T} \mathbb{E}_{q(\mathbf{z}_t^r|\mathbf{z}_0^r)} \|\mathbf{z}_0^r \odot M - d(\mathbf{z}_t^r, \text{``[V]''}, t; \theta \oplus \theta^r) \odot M\|^2. \quad (5)$$

During training, they use the prompt "[V]" instead of the original $P^r$, where [V] denotes a learnable token embedding for this reference subject. Moreover, they follow Ruiz et al. (2023) to use prior preservation loss, thus preventing the problems of language drift and reduced output diversity. The prior preservation loss is formulated as follows:

$$\min_{\theta^r, [V]} \mathcal{L}(\theta \oplus \theta^r, \mathbf{z}_0^{pr}, P^r), \quad (6)$$

where $\mathbf{z}_0^{pr}$ is the image latent of a pre-generated image by pre-trained SD using prompt $P^r$. Then the final loss is the sum of that in Equations 5 and 6. In particular, they adopt a two-phase training strategy, including solely optimizing the learnable token embedding, as well as joint learning of [V] and $\theta^r$. After training, with the SD parameterized by $\theta \oplus \theta^r$ and the learned token embedding of [V], we could use a new prompt containing [V] to customize this subject into various contexts.

## A.5 ADDITIONAL DETAILS ON DIFFUSION MODEL

**Forward Process.** Given a variance schedule $\beta_1, \ldots, \beta_T$ (*i.e.*, how much noise is added at each time-step), each $\mathbf{z}_t$ adheres to the following noisy sample distribution:

$$q(\mathbf{z}_t|\mathbf{z}_0) = \mathcal{N}(\mathbf{z}_t; \sqrt{\bar{\alpha}_t}\mathbf{z}_0, (1 - \bar{\alpha}_t)\mathbf{I}), \quad \text{where} \quad \alpha_t := 1 - \beta_t, \ \bar{\alpha}_t := \prod_{s=1}^{t} \alpha_s, \quad (7)$$

where the mean approaches 0 and variance approaches 1 as $T \to \infty$.

**Closed Form of** $q(\mathbf{z}_{t-1}|\mathbf{z}_t, \mathbf{z}_0)$**.** Given by $\mathcal{N}(\mathbf{z}_{t-1}|\tilde{\boldsymbol{\mu}}_t(\mathbf{z}_t, \mathbf{z}_0), \tilde{\beta}_t\mathbf{I})$, where

$$\tilde{\boldsymbol{\mu}}_t(\mathbf{z}_t, \mathbf{z}_0) = \frac{\sqrt{\bar{\alpha}_{t-1}}\beta_t}{1 - \bar{\alpha}_t}\mathbf{z}_0 + \frac{\sqrt{\alpha_t}(1 - \bar{\alpha}_{t-1})}{1 - \bar{\alpha}_t}\mathbf{z}_t, \ \tilde{\beta}_t = \frac{1 - \bar{\alpha}_{t-1}}{1 - \bar{\alpha}_t}\beta_t. \quad (8)$$

## A.6 ADDITIONAL RESULTS

**Time-step Coefficient Visualization.** In Figure 13, we visualize the learned time-step coefficients. The reference subjects in the first row consist of relatively small objects, while those in the second row represent a special case—backgrounds. Notably, the coefficients exhibit different trends for

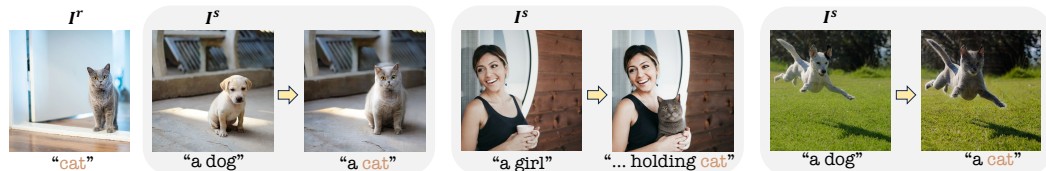

Figure 14: Customized image editing with significant changes of objects.

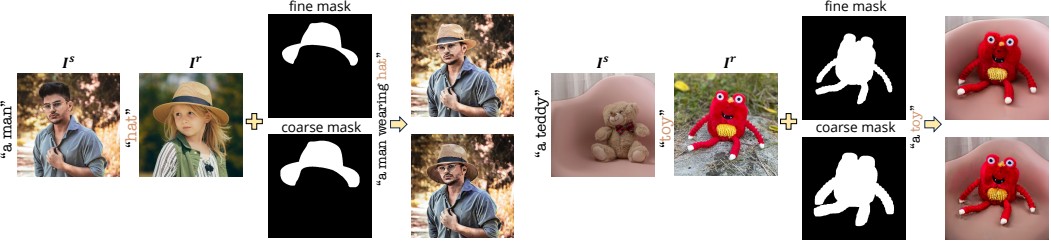

Figure 15: Customized image editing with fine and coarse masks of reference subjects.

customized image editing between these two types of reference subjects. Specifically, the coefficient values for reference backgrounds are generally higher than those for the source images, whereas the opposite trend is observed for small reference objects. Even among the same type of reference subjects, the coefficient values differ. For example, variations can be observed in the time-step visualizations for small reference objects and their corresponding source images in the first row of Figure 13. Consequently, the optimal time-step-specific coefficients, along with patch-specific coefficients, are distinct for different pairs of source and reference images to achieve the objective outlined in Equation 4. Moreover, the visualization examples broadly align with the relationship between the diffusion time-step and visual attributes, *i.e.*, using a large $\beta^s$ at a small time-step to preserve the fine-grained details in $I^s$ while increasing $\beta^r$ at a larger time-step to keep the more coarse-grained shape of the subject in $I^r$.

**Significant Object Changes.** Our TimeFusion supports significant changes of objects. In Figure 14, we provide the customized image editing results with the same reference object, *i.e.*, the cat. The position and posture of the cat could be adjusted according to the source images and text prompts, *e.g.*, from standing to sitting, being held, and jumping.

**Reference Subject Masks.** We use the approach in Avrahami et al. (2023) to learn the subject in a single reference image $I^r$. Avrahami et al. (2023) state that subject masks can be loose masks provided by the user, or generated by an automatic segmentation method (*e.g.*, SAM). We dilate the fine masks obtained by SAM to simulate coarse masks. As shown in Figure 15, even with coarse masks, we could generate comparable editing results.

**Additional Results.** Additionally, Figures 16 and 17 provide extra results for Figure 3, demonstrating the superiority of our TimeFusion for customized image editing.

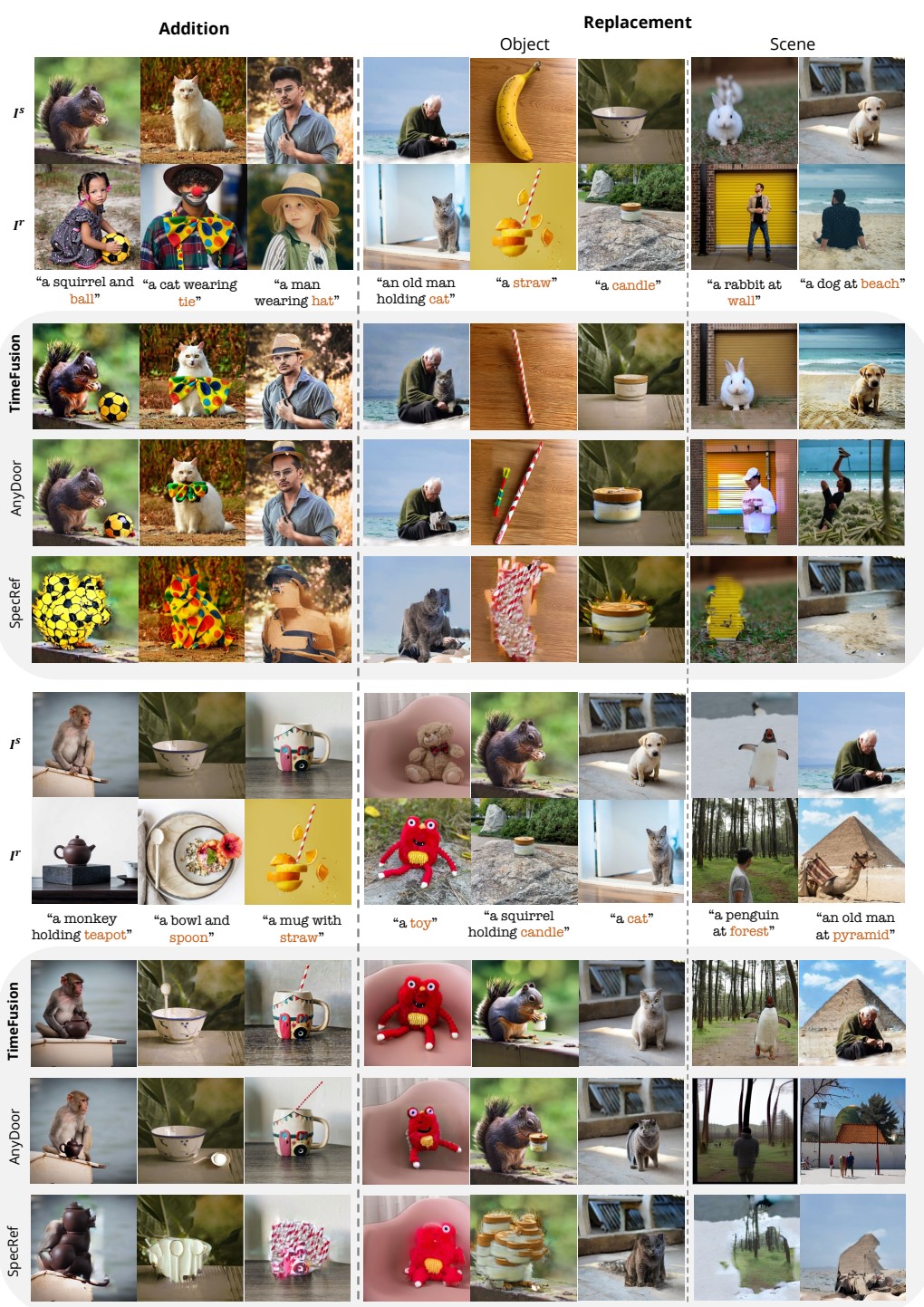

Figure 16: Comparison of customized image editing results between our TimeFusion and existing SOTAs Chen et al. (2023); Chen & Huang (2023). We compare 3 tasks, including addition, object and scene replacement (details in Section 3). The source prompt is omitted, and the reference subject prompt is highlighted in orange inside the target prompt. For fairness, examples are chosen based on their best visual quality from various random seeds. See Section 5 for analysis.

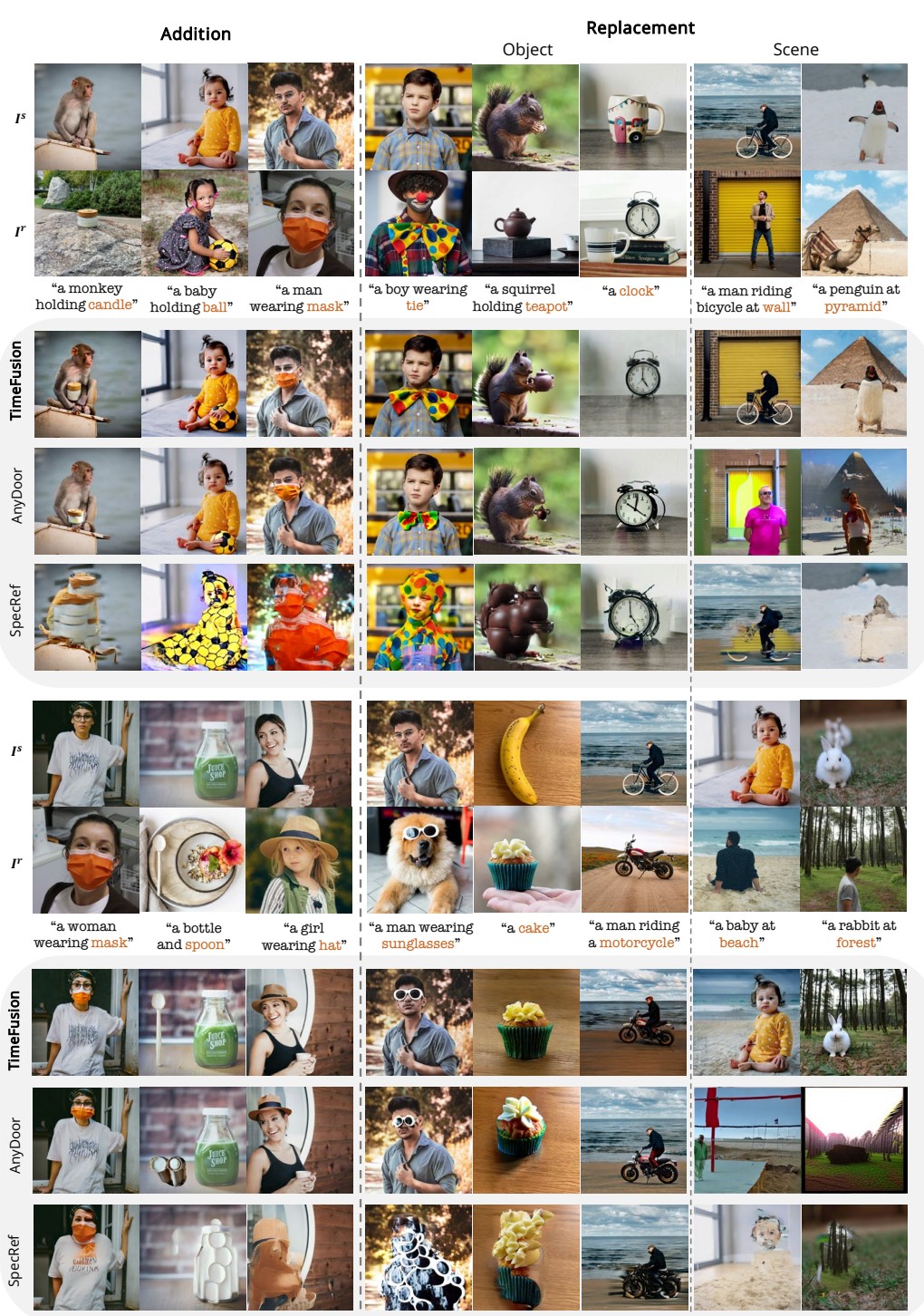

Figure 17: Comparison of customized image editing results between our TimeFusion and existing SOTAs Chen et al. (2023); Chen & Huang (2023). We compare 3 tasks, including addition, object and scene replacement (details in Section 3). The source prompt is omitted, and the reference subject prompt is highlighted in orange inside the target prompt. For fairness, examples are chosen based on their best visual quality from various random seeds. See Section 5 for analysis.

