# OpenReview forum: "Object Fusion via Diffusion Time-step for Customized Image Editing with Single Example"
_ICLR.cc/2025/Conference — ICLR 2025 Conference Withdrawn Submission_

### Official Review · Reviewer_EjFn · 2024-10-27

**Soundness:** 1
**Presentation:** 1
**Contribution:** 1
**Rating:** 3
**Confidence:** 5

**Summary:**

This paper tackles the challenging task of customized image editing with a text-conditioned DM, by improving the current LoRA fusion.

**Strengths:**

1. This paper focuses on a challenging task and design a method tailored to this task.
2. Some visual results look good.

**Weaknesses:**

1. The paper is not well-written. The abstract and introduction involve too many details. The introduction should mainly describe the motivation and high-level idea of the proposed method.
2. The discussion on the existing works is insufficient. Subject-driven image editing is a very broad topic and the authors should have a more thorough discussion. For example, object addition/insertion is also known as image composition, which should be discussed in introduction or related work. And more image compostion baselines should be compared.
3. The proposed method is trivial and lacks technical novelty.  The main novelty is time-specific and location-specific lora, which is also trivial and straightforward. Time-specific lora has been widely used in previous works. Since we need to combine source image and reference image, it is natural to use location-specific lora.
4. The user needs to specify the mask to be edited, which is laborious and troublesome. If the object shape needs to be slightly changed in the source image, how to address this issue?
5. The proposed method needs to optimize the fusion coefficients for each pair of source image and reference image, which is very tedious. What about the time cost of three steps? There is not efficiency comparision between the proposed method and baselines. Is the proposed method more time-consuming than the baselines? If we change a reference object, it is necessary to optimize the fusion coefficients again?
6. In the experiments, the comparision with existing methods is not adequate and more related methods should be compared. Moreover, the authors conduct experiments on the self-collected datasets, so the experimental results are not convincing.
7. For the evaluation metrics, the authors only report the similarity metrics (e.g., CLIP, DINO) between the inserted/replaced object and the reference object. However, the quality of the whole image should be measured from multiple aspects, for example, whether the inserted object is compatible with the background in terms of illumination, viewpoint, and so on. Therefore, the evaluation is not convincing.
8. There is no discussion on the failure cases. What about the ability scope of the proposed method? Based on my understanding, the proposed method has no ability to adjust the view or pose of inserted object. If so, the ability of proposed method is very limited. There is no showcase for the challenging cases.

**Questions:**

See the weakness.

---

### Official Review · Reviewer_EVGe · 2024-11-04

**Soundness:** 3
**Presentation:** 3
**Contribution:** 2
**Rating:** 5
**Confidence:** 4

**Summary:**

The task of this paper is image editing. Specifically, given a source image and a reference image with a target object, the goal is fusing the reference object into the secure image. The major challenge is how to perform editing, while maintaining the fidelity of both source and reference.

This method is build upon the line of works that utilize diffusion model and lora fine-tuning for customized editing.  Instead of naively learning two LoRAs for source and target reconstruction and combine them with a constant weight, the author proposes to learn a time-variant weighting. They also use patch coefficients for better editing localization.

**Strengths:**

The motivation of the task is clear, and using a time variant weight for source and target make sense.

The visualisation of time weight is interesting and it strengthen the motivation.

There are both quantitative and qualitative comparisons, which do show the method can achieve decent editing.

**Weaknesses:**

1. Although the motivation is that time variant weighting is important, the actually weight is learned. Can one  design a rule and just vary the weighting during sampling? i.e. t is large for ref during beginning and slowly decrease? It will be nice to show such results.
2. No quantitative results for ZipLORA, Lora Switch etc.
3. It would be beneficial to see why using reconstruction objective can lead to good editing sampling.

**Questions:**

1. This method learns only to reconstruct source and target object. By using the objective and use the learned lora during sampling, they magically performs editing well. However, reconstruction objective could be tied to spatial locations, and I wonder how robust the method is if the reference object location is very different than the desired location after editing.For most showed result, the location variance is little.
2. Instead of learning in two stage, what will happen if lean in one stage? That’s more computationally efficient.

---

### Official Review · Reviewer_PmP6 · 2024-11-04

**Soundness:** 2
**Presentation:** 2
**Contribution:** 2
**Rating:** 5
**Confidence:** 4

**Summary:**

In this paper, the authors propose TimeFusion, a LoRA-based model that performs object fusion through a time-aware weighting mechanism. A global weight is initially used to integrate the well-trained target and reference LoRAs to help maintain object fidelity. Additionally, a spatial weight is employed to further refine the object fusion process. The authors also provide an internal analysis to verify the effectiveness of this fusion strategy.

**Strengths:**

- A time-aware fusion strategy is proposed to enhance LoRA fusion, ensuring that the results maintain high fidelity.
- Extensive experiments and results demonstrate the validity of this fusion approach.

**Weaknesses:**

- The proposed TimeFusion seems more like a trick than a legitimate paper or piece of work. First, it can only handle object fusion by training object-centric LORAs and combining them, which is too limited. In contrast, previous LORA techniques can manage both object and style fusion. Additionally, there are numerous methods focused on object generation and editing through inpainting, which can also achieve visually pleasing results. Given this perspective, I am uncertain why the authors emphasize LORAs for this task. Second, there are many follow-ups to Anydoor that maintain fidelity well, such as CustomNet[1] and IMPRINT[2]. I wonder why the authors did not compare their work to these methods. Third, from my perspective, this work essentially performs a spatially aligned copy-and-paste from the reference image to the target image. For example, in Fig. 3, the ‘clock’ in the target image nearly occupies the same spatial position as in the reference image.
- I doubt the possibility of combining multiple LORAs together, either in the case of spatial alignment or spatial misalignment.

[1] Multi-LoRA Composition for Image Generation

[2] CustomNet: Zero-Shot Object Customization with Variable-Viewpoints in Text-to-Image Diffusion Models

[3] IMPRINT: Generative Object Compositing by Learning Identity-Preserving Representation

**Questions:**

- In Figure 3, why do your results appear to have background colors that vary greatly, such as the wall, teapot, and monkey?
- How would your model perform if you did not use a mask when optimizing the target LORA?

---

### Official Review · Reviewer_J8P4 · 2024-11-05

**Soundness:** 3
**Presentation:** 3
**Contribution:** 3
**Rating:** 6
**Confidence:** 3

**Summary:**

The paper "Object Fusion via Diffusion Time-Step for Customized Image Editing with Single Example" presents TimeFusion, a novel method to improve customized image editing. This is done by using a text-conditioned Diffusion Model. The goal of the proposed method is to merge elements from reference and source images without losing the visual quality of either. The authors suggest that existing methods, such as LoRA fusion, have difficulty maintaining a balance between the quality of the reference and source images. TimeFusion solves this problem by introducing specific LoRA fusion coefficients for each time-step, allowing for more precise control over the detailed and broad attributes of the fused images. The paper provides both qualitative and quantitative experiments that show TimeFusion performs better than current leading methods.

**Strengths:**

1. The uniqueness of the method comes from its use of coefficients specific to each time-step and patch, which provides precise control over image fusion.
2. TimeFusion shows excellent ability in maintaining the quality of both source and reference attributes across various editing tasks, performing better than current leading methods.
3. The paper is backed by a thorough experimental setup, clear visuals, and both numerical and visual evaluations.
4. TimeFusion adapts well to different prompts, demonstrating its wide-ranging applicability in real-world scenarios.

**Weaknesses:**

1. The experiments primarily concentrate on basic tasks such as object addition or replacement, and do not include complex scenarios like dynamic backgrounds or scenes with multiple overlapping objects. This could limit the applicability of the results in more complex situations.
2. Although the computational requirements are discussed, there is no analysis of their practical implications. For example, it's unclear whether the computational load could prevent the method from being used in real-time or large-scale applications.
3. The extension from existing LoRA methods is not fundamentally new, as it mainly modifies the fusion process. The introduction of time-step-specific coefficients is a minor improvement and may not warrant a separate method unless its broader applicability is demonstrated.

**Questions:**

1. Could you explain the rationale behind grouping time-steps into 20 splits for coefficient learning? Have you investigated other grouping configurations to determine if there might be a more effective approach?

2. How does TimeFusion handle real-world challenges such as varying lighting conditions, object occlusions, or multiple subjects in a reference image?

3. Considering the computational demands of your method, what are the prospects for scaling this approach for practical applications, such as mobile devices or real-time editing software?

---

### Note · Authors · 2024-11-15

I have read and agree with the venue's withdrawal policy on behalf of myself and my co-authors.